# Impact of Obesity and Bariatric Surgery on Metabolic Enzymes and P-Glycoprotein Activity Using the Geneva Cocktail Approach

**DOI:** 10.3390/jpm13071042

**Published:** 2023-06-25

**Authors:** Hengameh Ghasim, Mohammadreza Rouini, Saeed Safari, Farnoosh Larti, Mohammadreza Khoshayand, Kheirollah Gholami, Navid Neyshaburinezhad, Yvonne Gloor, Youssef Daali, Yalda H. Ardakani

**Affiliations:** 1Biopharmaceutics and Pharmacokinetic Division, Department of Pharmaceutics, Faculty of Pharmacy, Tehran University of Medical Sciences, Tehran 1417614411, Iran; 2Department of General Surgery, Firoozgar General Hospital, Iran University of Medical Sciences, Tehran 1417614411, Iran; 3Department of Cardiology, Imam Khomeini Hospital Complex, Tehran University of Medical Sciences, Tehran 1417614411, Iran; 4Department of Drug and Food Control, Faculty of Pharmacy, Tehran University of Medical Sciences, Tehran 1417614411, Iran; 5Department of Clinical Pharmacy, Faculty of Pharmacy, Tehran University of Medical Sciences, Tehran 1417614411, Iran; 6Division of Clinical Pharmacology and Toxicology, Geneva University Hospitals, 1205 Geneva, Switzerland; yvonne.gloor@unige.ch

**Keywords:** phenoconversion, bariatric surgery, CYP450 phenotype and genotype, cocktail, personalized medicine

## Abstract

The inter-individual variability of CYP450s enzyme activity may be reduced by comparing the effects of bariatric surgery on CYP-mediated drug elimination in comparable patients before and after surgery. The current research will use a low-dose phenotyping cocktail to simultaneously evaluate the activities of six CYP isoforms and P-gp. The results showed that following weight reduction after surgery, the activity of all enzymes increased compared to the obese period, which was statistically significant in the case of CYP3A, CYP2B6, CYP2C9, and CYP1A2. Furthermore, the activity of P-gp after surgery decreased without reaching a statistical significance (*p*-value > 0.05). Obese individuals had decreased CYP3A and CYP2D6 activity compared with the control group, although only CYP3A was statistically important. In addition, there was a trend toward increased activity for CYP1A2, CYP2B6, CYP2C9, and CYP2C19 in obese patients compared to the control group, without reaching statistical insignificance (*p*-value ≥ 0.05). After six months (at least), all enzymes and the P-gp pump activity were significantly higher than the control group except for CYP2D6. Ultimately, a greater comprehension of phenoconversion can aid in altering the patient’s treatment. Further studies are required to confirm the changes in the metabolic ratios of probes after bariatric surgery to demonstrate the findings’ clinical application. As a result, the effects of inflammation-induced phenoconversion on medication metabolism may differ greatly across persons and drug CYP pathways. It is essential to apply these results to the clinic to recommend dose adjustments.

## 1. Introduction

Phenoconversion signifies the mismatch between an individual’s genotype-based predicted capacity for drug metabolism and actual capability due to non-genetic factors. Phenoconversion can be caused by both extrinsic and patient- and disease-related factors. It is, however, unknown what causes this occurrence and the extent to which CYP450 metabolism is impacted. Co-morbidities such as aging, cancer, inflammation, and the concurrent utilization of CYP450-inhibiting drugs have all been linked to phenoconversion into a lower metabolizer phenotype. Concomitant usage of CYP450 inducers and smoking has been linked to phenoconversion into a higher metabolized phenotype. The research investigations reported genotype–phenotype discrepancies; however, the impact of phenoconversion on clinical efficacy and toxicity remains unknown. If phenoconversion is investigated in greater detail, it could enhance our capacity to predict individual CYP450 metabolism and personalize drug treatment [1,2]. Before implementing “personalized medicine” in clinical practice, numerous issues arising from a genotype-based analysis of relevant DMEs must be resolved. The patient is at great risk from phenoconversion caused by infection and other inflammatory conditions. As a result, a patient’s genotype provides only a small percentage of the data needed to support the concept of “the right drug at the right dose the first time” [3].

The findings suggest that high levels of various pro-inflammatory cytokines developed during inflammation are down-regulating drug metabolism, particularly by some DMEs of the P450 family, probably inducing transient phenoconversion. The investigation supports that certain inflammatory situations linked with higher pro-inflammatory cytokines could promote DME phenoconversion. Since the significant prevalence of inflammatory conditions is linked to increased levels of pro-inflammatory cytokines, phenoconversion of genotypic EM patients into temporary phenotypic PMs may occur more often than is generally recognized. Given that drug pharmacokinetics, and, consequently, clinical response, is affected by DME phenotype rather than genotype per se, phenoconversion may compromise the future of personalized medicine [4,5,6].

Obesity is becoming increasingly common, with over 1.9 billion individuals worldwide estimated to be overweight or obese. Conditions including hypertension, type 2 diabetes, cardiovascular disease, nonalcoholic fatty liver disease, asthma, and cancer are all linked to obesity and have a high mortality and comorbidity rate. Patients who are overweight often have to take medication for a longer period than the average person. It is generally agreed that bariatric surgery is the best effective therapy for extreme obesity at present [7,8,9]. The BMI index, displayed in Table 1, serves as the classification’s foundation.

Sleeve gastrectomy (SG) is used in most bariatric surgeries performed globally. Roux-en-Y gastric bypass (RYGB): This procedure entails creating a tubular stomach and restructuring the residual stomach into a tube or sleeve-like shape. It minimizes the stomach to the size of a 30 cc tiny gastric pouch surgically connected to the jejunum. Most of the stomach and duodenum are stapled and bypassed in this procedure, as seen in Figure 1 [11,12]. LRYGB may modify post-operative drug pharmacokinetics (due to the anatomical and physiological changes), which can lead to adverse effects or vary their responses in patients, necessitating dosage modification in these individuals [7].

A persistent, low-grade form of inflammation is obesity. Adipocytes increase, and macrophages enter the growing adipose tissue as an individual’s weight gain or obesity progresses. Numerous pro-inflammatory cytokines and adipokines, notably interleukin-6 (IL-6), tumor necrosis factor-a (TNF-α, and leptin, may be raised by abdominal or visceral obesity. Less attention has been paid to how adipokines or their interactions with cytokines may alter other key aspects of medication disposition in obesity. It has previously been established that pro-inflammatory cytokines may coexist with a reduced expression of particular drugs metabolizing enzymes and drug transport proteins [13]. Cytokines are small, cell-secreted proteins that directly impact cell interactions and communications. Interleukins are also cytokines produced by one cell that influence other leukocytes. Some cytokines are both pro- and anti-inflammatory. The central sensitization caused by nerve injury or inflammation is also mediated by specific inflammatory cytokines [14]. According to in vitro studies, DMEs from the CYP1, CYP2, and CYP3 families may be suppressed directly by inflammatory stimuli. The strongest evidence for this suppressive impact has been discovered for IL-6, IL-1, TNF-α, and LPS [15].

The main human enzyme system that metabolizes drugs is called the cytochrome P450 (CYP) enzyme. Environmental factors or genetic variations may influence these enzymes’ activity. Along with metabolizing enzymes, drug inflow and efflux proteins, such as P-glycoprotein (P-gp), play a crucial role in the pharmacokinetic variability of drug response. CYP and/or P-gp activity pharmacokinetic changes may have various toxicological and pharmacological consequences. As a result, it is crucial to precisely evaluate their in vivo activity (phenotyping). By using a cocktail technique that comprises the injection of many CYP- or P-gp-specific probe agents, it can simultaneously analyze the effects of these enzymes and the transporter [16,17,18].

The CYP probes in the Geneva cocktail were not shown to interact with one another in prior studies. This further verifies the cocktail’s ability to predict normal and modified CYP activity accurately. Due to the combination of a straightforward, user-friendly capillary sample device [19], this low-dose, high-throughput concoction can be utilized as a phenotyping instrument in various settings, including hospitals, private clinical practice, and research in countries with limited resources.

## 2. Materials and Methods

### 2.1. Study Design and Population

To identify a >30% reversal in CYP3A activity with 80% power and a 5% significance level, a sample size of 16 individuals was necessary. A sample size of 20 healthy, non-obese participants was consequently selected as the control group. A sample size of 24 people was intended for obese patients to prevent loss to follow-up and consider a coefficient of 0.55 as statistically significant with 80% power and a value of 5%. Given that the standard deviation (SD) of the differences is 36% (estimated MR standard deviation for CYP3A relying on the literature), the sample size of 24 participants allows the detection of a >22% variation in CYP MRs between pairs. A *p*-value < 0.05 was regarded as statistically significant. Figure 2 provides a summary of the research’s design. The protocol was previously published in detail [20]; however, some basic points are mentioned below (Figure 3).

Participants were sought for an open-label, non-randomized, two-arm study comparing the metabolic effects of obesity on the activity of six cytochromes P450 enzymes and P-gp, and the effect of bariatric surgery on alterations to the phenotypic pattern of enzymes, or phenoconversion. The investigation’s recruitment period lasted from April 2019 to June 2021. All experiments and clinical studies were conducted following the guidelines established by Basic & Clinical Pharmacology & Toxicology [21]. The following were the inclusion criteria [22] for the investigation:Adolescent volunteers having a BMI of <25 and severely obese patients with a BMI of ≥40 who are candidates for bariatric surgery;Between the ages of 18 and 60;HbA1C < 6.5%;All healthy and obese individuals alike must not have active cases of thyroid illness, cirrhosis, IBD, Helicobacter pylori infection, or any other infectious disease (now or during the last four weeks).

In addition, the exclusion criteria were as follows:


Participants taking medications believed to influence metabolic activity, including corticosteroids or NSAIDs used for their anti-inflammatory effects, will not be allowed to participate in this study;Females will be questioned regarding having a normal menstrual cycle and will be disqualified if they are pregnant or breastfeeding;Individuals who are often treated with drugs that have an inhibitory or stimulating impact on DMEs and whose substrates are implemented in the phenotyping cocktail;Patients who have previously had hypersensitivity to any of the drugs included in the combination;Patients who had organ transplant surgeries;Patients with an active cancer;Participants who had been heavy smokers or alcoholics for at least two months before the research.


The potential risks of the research were made known to each participant. Before beginning any research procedures, each subject was allowed to provide written informed consent. All participants had interviews to discover more about their disease history and present drug treatment. Participants in the research were instructed not to drink alcohol within 48 h and to abstain from excessive activity and caffeine consumption for 24 h before the visits. We are sure that, considering these circumstances, a competitive inhibition between their usual prescriptions and the pharmaceuticals in the cocktail is improbable. The administration of medications was prohibited on the morning of these appointments. The average first visit was 17 days (1 to 218 days) before the procedure. The second visit happened at a median of 189 days (178–678 days) following surgery, when both a significant and consistent weight loss had been achieved. Table 2 compiles the research findings of the different reference groups of healthy participants.

### 2.2. Ethical Approval

The Tehran University of Medical Sciences Ethics Review Board approved the research protocol (ID: IR.TUMS.TIPS.REC.1397.104). The process met the ethical standards established by the national and/or institutional research committee.

### 2.3. Cocktail Administration

During visits 1 and 2 of the study, patients were administered the following doses of the oral drug cocktail. The cocktail was administered in two capsules made of equally laden cellulose. Tablets were pulverized in a local laboratory pharmacy to create the cocktail. It contained 100 milligrams of caffeine (CYP1A2) (sachet, Multi Cafè 2in1, Tehran, Iran), 25 mg of bupropion (CYP2B6) (tablet, Abidi, Tehran, Iran), 25 mg of flurbiprofen (CYP2C9) (tablet, Sanovel, Istanbul, Turkey), 5 mg of omeprazole (CYP2C19) (granules in the capsule, Abidi, Tehran, Iran), 10 mg of dextromethorphan (CYP2D6) (powder, Alhavi, Tehran, Iran), 1 mg of midazolam (CYP3A4/5) (liquid formulation, Darou Pakhsh, Tehran, Iran), and 25 mg of fexofenadine (P-gp) (tablet, Abidi, Tehran, Iran).

Caffeine and midazolam were given separately as oral solutions immediately following capsule administration. Since heparin has been shown to partly hinder PCR [23], venous blood samples were taken into non-heparinized tubes at the following time points: baseline, 1, 2, and 3 h after cocktail administration. Afterward, blood samples were centrifuged at 15,000× *g* for 10 min. For subsequent analysis, the plasma samples were stored frozen at −80 °C.

### 2.4. Laboratory Sample Analysis

A recently approved and developed liquid-chromatography tandem mass-spectrometric technique was used to evaluate the concentrations of probe drugs and their associated metabolites in plasma samples [24,25]. MRs (metabolic ratio of metabolite to parent drug) at 2 (CYP1A2 and CYP3A) and 3 h (CYP2B6, CYP2C9, CYP2C19, and CYP2D6) could be used as phenotyping measures to assess CYP activity. In addition, the phenotypic indicator for P-gp is AUC0-3 (area under the plasma concentration–time curve between 0 and 3 h after administration).

ELISA Kit measured pro-inflammatory cytokine levels (IL-1β and IL-6). A single operator assessed stroke volume (SV) and cardiac output (CO) during an echocardiogram for all patients in the echocardiography section of the Imam Khomeini Hospital complex. The quantities were normalized for both parameters by dividing them by Body Surface Area (BSA).

The research used full blood samples from patient and volunteer groups to extract genomic DNA. According to other research [25,26,27], DNA samples were employed for genetic analyses related to the specified CYP450s alleles using RT-PCR (TaqMan assay). The phase-predicted metabolized status is used throughout the accompanying text to refer to genotype-predicted group assignment.

### 2.5. Data Management

A unique identifier was assigned to each of them to protect the participants’ privacy and facilitate their identification. The code and participant data were only accessible to investigators. Throughout the entire study process, the data were categorized in coded files. The samples were only employed for this investigation. Genotype, activity phenotype, and clinical data were acquired independently, and the technicians on each side were blinded until all data sets were statistically ready [28].

### 2.6. Statistical Analysis

The Shapiro–Wilk Test was employed to determine the normality of the quantitative data. The population’s characteristics throughout time were displayed as mean ± standard deviation. The *t*-test or paired *t*-test was employed to evaluate quantitative data relating to the phenotype for each study group (comparing persons before and after treatment with the healthy normal group or assessing patients’ status (before and after treatment) about themselves). The statistical evaluations were carried out using IBM SPSS Statistics 26. All significance thresholds were set at 5% and were two sided. A paired analysis was carried out between the pre-operative and post-operative periods, and an unpaired analysis was performed between the healthy and patient groups.

### 2.7. Safety

Midazolam demonstrated a modest sedative pharmacological impact, but no additional side effects were noted.

## 3. Results

### 3.1. Demographic and Paraclinical Results

A total of 24 patients, with a mean age of 36.6 years and a gender distribution of 21 females and 3 males, were selected from the surgeon’s patient group. As a result, they underwent evaluation in two sessions separated by at least six months. Additionally, only one sample was taken from 21 healthy, non-obese volunteers with a mean age of 30, consisting of 14 males and 7 women. Age, sex, height, weight, BMI, HbA1c, FBS, AST, ALT, CO, SV, IL-1β, and IL-6 were some of the demographic and paraclinical traits examined between obese and non-obese healthy volunteers. The fact that these factors were evaluated in OBESE patients before and after the six-month therapy period must be emphasized. Table 2 presents the results of these variables and statistical comparisons.

NA, not applicable; BMI, body mass index; HbA1c, glycated hemoglobin; DPP4-I, dipeptidyl peptidase-4 inhibitors; ACEI, angiotensin-converting enzyme inhibitors; ARB, angiotensin II receptor blockers; CCB, calcium channel blockers; NSAID, non-steroidal anti-inflammatory drugs; PPI, proton pump inhibitors, OCP, oral contraceptive pills.

The average weight lost overall six months following surgery (visit 2) was 27.9%. After surgery, most biological markers, particularly liver transaminases, developed.

### 3.2. The Echocardiography Results of the Patients before and after Surgery

The echocardiography process in patients demonstrated that the index of SV and CO was considerably higher after surgery (40.6 ± 5.7 and 2788 ± 663 cc/m^2^, respectively) in comparison to before surgery (31 ± 5.7 and 2425 ± 475 cc/m^2^, respectively).

### 3.3. The Pro-Inflammatory Cytokines Level of the Patients before and after Surgery and Healthy Group

When compared to healthy volunteers (the control group), obese patients had higher levels of inflammation (as determined by IL-1β and IL-6 levels), with IL-1β and IL-6 levels of 2.1 ± 3.1 and 7.3 ± 10.1 pg/mL and 0.9 ± 0.6, and 2.5 ± 1.5 pg/mL, accordingly. Patients’ inflammation levels were also considerably higher following bariatric surgery than pre-surgery (3.1 ± 5.2 vs. 9.6 ± 11.0 pg/mL, *p* < 0.05).

### 3.4. CYP450 Genotype of the Study Population

We examined the genotypes of healthy individuals and obese patients during the phenotyping of the enzymes to exclude those with a poor metabolizer (PM) genotype. Table 3 indicates the frequency of the genotypes’ anticipated activities.

Figure 4 also illustrates the metabolic ratio, which compares healthy individuals’ actual activity with patients’ projected activity depending on their genotype.

### 3.5. Phenotype Results

We examined the activity of CYP1A2 and CYP3A4/5 enzymes 2 h after drug cocktail administration and the activity of CYP2B6, CYP2C9, CYP2C19, and CYP2D6 enzymes 3 h after drug cocktail administration relying on their metabolic ratio, as well as the level of fexofenadine probe 0–3 h (AUC0-3) after drug cocktail administration to evaluate P-gp pump activity.

The results showed that following weight reduction after surgery, the activity of all enzymes increased compared to the obese period, which was statistically significant in CYP3A, CYP2B6, CYP2C9, and CYP1A2. Furthermore, the activity of P-gp after surgery decreased without reaching a statistical significance (*p*-value > 0.05). Obese individuals had decreased CYP3A and CYP2D6 activity compared with the control group, although only CYP3A was statistically significant. Additionally, obese individuals tended to display increased activity for CYP1A2, CYP2B6, CYP2C9, and CYP2C19 compared to the control group. However, this trend did not statistically differ from that of the control group (*p*-value ≥ 0.05). All enzymes and the P-gp pump had considerably greater activity than the control group after at least six months, except for CYP2D6.

### 3.6. CYP1A2

Before surgery, participants in excellent health had metabolic ratios that were higher (0.129 ± 0.073) than those of obese patients (0.1660 ± 0.145), although this difference was not considered significant (*p* > 0.05). This ratio measures the activity of the CYP1A2 enzyme.

Additionally, there was a significant difference between before and after surgery (*p* < 0.05), and patients’ activity increased significantly (0.341 ± 0.396) after surgery compared to the control group (Figure 5).

### 3.7. CYP2B6

The findings demonstrated that, while not statistically significant, the metabolic ratios were greater in the obese patients before surgery (3.954 ± 2.257) compared to the control group (2.855 ± 0.988). The metabolic ratios significantly enhanced after surgery (5.969 ± 3.703) (*p* < 0.05) compared to the control group and the obese patients before surgery (Table 4 and Figure 6).

### 3.8. CYP2C9

The activity of CYP2C9 improved after surgery (0.110 ± 0.047), and this difference between patients before and after surgery and in comparison to the control group was statistically significant (*p* < 0.05; Table 4 and Figure 7). The mean metabolic ratio and CYP2C9 activity before surgery were slightly higher than the control group (0.074 ± 0.019), but this difference was not statistically significant (*p* > 0.05).

### 3.9. CYP2C19

Before surgery, obese patients had greater metabolic ratios and CYP2C19 enzyme activity (1.683 ± 2.767) than the control group (0.686 ± 0.558), but there was no statistically significant difference (*p* > 0.05) (Table 4 and Figure 8). The activity of this enzyme substantially enhanced after surgery (1.887 ± 2.003) compared to the control group (*p* < 0.05); however, the difference between the individuals before and after surgery was statistically insignificant (*p* > 0.05).

### 3.10. CYP2D6

The dextromethorphan metabolic ratio, which represents the activity of the CYP2D6 enzyme, differed slightly but not significantly between participants before (1.091 ± 1.082) and after (1.761 ± 1.592) surgery in comparison to the control group (1.242 ± 0.835), as shown in Table 4 and Figure 9.

### 3.11. CYP3A4/5

The findings showed that the metabolic ratio and CYP3A4/5 enzyme activity in obese patients were significantly lower before surgery (0.419 ± 0.257) than in the control group (0.633 ± 0.253) (*p* < 0.05) but significantly higher after surgery (1.000 ± 0.590) than in the control group (*p* < 0.05). Table 4 and Figure 10 show the significant difference (*p* < 0.05).

### 3.12. P-gp Pump

Before surgery, obese patients’ plasma concentrations of the fexofenadine probe (AUC0-3), which measures P-gp pump activity, were significantly lower in the obese patients’ group (99.9 ± 98.7) than in the control group (248 ± 91) (*p* < 0.05). This is because the two variables have an inverse relationship. The concentration rose after surgery (143.1 ± 174.4), with no significant change between before and after surgery (*p* > 0.05); however, it remained considerably lower than the control group (*p* < 0.05) (Table 5 and Figure 11).

## 4. Discussion

Cytochromes from the P450 family (CYPs) are key players in the pharmacokinetics of pharmaceuticals, either by catalyzing pro-drug conversion into active components or by facilitating their excretion. The CYP family of enzymes is hypothesized to exhibit extensive inter- and intra-individual variability, driven by genetic and environmental factors such as drug–drug interactions, diseases, inflammatory state, smoking, or nutritional intake [29]. It has been verified that variations in liver cytochrome activity affect the effectiveness and safety of the therapy. While the proper dosage is crucial for drugs with limited therapeutic range ineffectiveness, the emergence of unfavorable effects may also significantly influence treatment adherence and success. The Clinical Pharmacogenetics Implementation Consortium (CPIC) has previously released several recommendations for CYP genetic-based dose modifications for multiple drugs [30,31,32,33,34]. According to the nonclinical data compiled earlier, elevated levels of pro-inflammatory cytokines, especially IL-6, seen throughout an infection or inflammation downregulate drug metabolism, a characteristic of clinical phenoconversion of the DME from EM to PM phenotype, changing the pharmacokinetics of the substrate drugs and, ultimately, modifying the drug response (either therapeutic effectiveness or undesirable toxicity), regardless of patient’s EM genotype [4,35].

According to the activity of their cytochromes, patients are often classified as either ultra-rapid (UM), normal (NM), occasionally intermediate (IM), or poor (PM) metabolizers. Methods for predicting and evaluating a patient’s metabolizer status have evolved using genotyping and phenotyping. Phenotyping provides an overview of the actual metabolizing state at the moment of testing by combining genetics and all environmental elements. Genotyping indicates the inherent feature of the patient’s capacity to produce an active enzyme. Phenoconversion is the difference between a patient’s genotype-based anticipated phenotype and metabolizer status discovered during phenotyping. While this phenomenon is unlikely to greatly influence overall risk evaluation in a worldwide population, the elevated risk of adverse medication responses and unfavorable outcomes in certain subgroups of patients with a high incidence of phenoconversion may become important. Personalization of regular dosage regimens involving the patients’ genuine metabolizer status may become cost efficient in certain instances [30].

Tandra et al. used a cocktail technique to compare the action of CYP1A2, CYP2C9, CYP2C19, and CYP3A4 in RYGB patients with healthy control participants [36]. In this work, we employed a unique combination of seven medications to investigate modifications in their pharmacokinetics after bariatric surgery, which might demonstrate changes in CYP activity in each patient individually. The metabolite/parent drug ratio best indicates the patient’s corresponding exposure to the parent medication and its metabolite. We also calculated the AUC0-3 of the main metabolites and CYP probes.

Bariatric surgery had the significant largest effect on the metabolic ratios of midazolam, caffeine, and bupropion, which were considered as CYP3A (2.3-fold), CYP1A2 (2.0-fold), and CYP2B6 (1.5-fold) probes, respectively, as was reported similarly by Puris et al. in the case of CYP1A2 [7]. The CYP2C9 metabolic ratio was increased (19.6%), which was statistically significant, while Lloret-Linares et al. showed a transitory and significant increase in the CYP3A4/A5 and CYP2C9 metabolic ratios immediately after surgery [37]. Several drugs, including immunosuppressants (such as tacrolimus and cyclosporine), anti-vitamin K drugs (such as acenocoumarol, warfarin, and fluindione), anti-arrhythmic drugs (such as disopyramide and amiodarone), statins, and others, used in this situation increase the risk of reduced drug exposure. Theoretically, persons who use medications that are significantly metabolized by these CYPs should also be closely monitored during the first few days after surgery.

The notion that greater cytokine levels lead to reduced CYP450 activity, resulting in poorer drug metabolism and higher drug plasma, is supported by the observation that alterations in metabolic ratios of midazolam were “converted” by surgery to be equivalent to what occurred in the reference group (Figure 10).

According to de Jong et al., a little variation was seen in the dextromethorphan metabolic ratio (CYP2D6 substrate), suggesting that the levels of CYP2D6 MR are not influenced by obesity/bariatric surgery. In their investigation, CYP2D6 was determined to be the least susceptible to inflammation, which may be because nuclear receptors cannot activate it and are hence immune to changes in inflammation-induced PXR, CAR, and AhR levels that control the expression of other CYPs [15]. 

Fexofenadine, a validated P-gp pump substrate, had a 43% rise in AUC0-3 following bariatric surgery, and its AUC0-3 value was less than half that of healthy volunteers before the procedure. Consequently, it is intriguing that P-gp activity is greater in obese patients than in healthy volunteers. Still, previous research found that type 2 diabetes patients had reduced activity relative to healthy people [38]. The findings support research by Lenoir et al. that found P-gp phenotypic activity, not P-gp polymorphisms, has a meaningful influence on exposure to apixaban and rivaroxaban [39]. Omeprazole MR was somewhat higher (12.1%) in patients, yet this difference did not achieve statistical significance. When the AUC of the parent drug was investigated, they not only did not decrease after surgery (as would be projected due to the reduction in the absorption site), but they also increased substantially in patients who experienced bypass surgery (because the pH is higher in this type of surgery rather than sleeve surgery [40]). These findings are probably attributable to the elevated AUC of enteric-coated omeprazole granules at the new pH [7].

As documented by Jain et al. [10], accelerated gastric emptying, elevated cardiac output, and variations in enterohepatic recirculation in obesity may result in alterations in drug absorption. For instance, decreased gastric volume causes an elevation in stomach pH (4–6) and a change in gastric emptying time, which may influence the pace and amount of oral absorption. The absorption surface is decreased when the duodenum is bypassed, and intestinal transport and first-pass metabolism, controlled by intestinal CYP enzymes, may change. In the end, weight loss may normalize liver fat levels, improve hepatic insulin sensitivity, and reduce low-grade inflammation, which could alter hepatic clearance and drug absorption [7].

The process of disintegration for the oral administration of solid dosage forms is essentially formulation specific. Due to the smaller volume of the shortened stomach, RYGB attenuates gastric mixing, which is essential to the disintegration process (Figure 2). Therefore, anticipated formulations may exhibit a reduced disintegration rate. Drugs are categorized using the biopharmaceutical classification system based on their solubility and permeability properties. Due to its isolation from other acid-producing cells in the more distal regions of the stomach, the tiny gastric pouch that develops after RYGB elevates the pH of the stomach. Drugs that need an acidic environment for the best breakdown, such as rifampin, digoxin, and ketoconazole, seem more vulnerable to elevated stomach pH after RYGB. It is projected that variations in short intestine transit durations would affect the absorption of poorly soluble or extended-release pharmaceutical formulations. Time passing through the RYGB and intestinal tract may be reduced by avoiding the duodenum and proximal jejunum; however, insufficient transit time may hinder poorly water-soluble medications and extended-release formulations from fully dissolving and absorbing [13]. However, it is doubtful that these changes in the pharmacokinetics of drugs would influence patients’ health.

According to the results of our study, obesity-related moderate chronic inflammation affects CYP450 activity differently depending on the isoform. We contend that the inter-subject variation in medication response seen in obese individuals using prodrugs, drugs connected to toxicity, may be primarily caused by the influence of obesity on drug-metabolizing enzymes [26].

The clinical result of pharmaceutical treatments might vary widely across people and even within the same person. As a result, some individuals may suffer decreased effectiveness or an increased chance of experiencing adverse effects. While genetic variants in drug-metabolizing enzymes (DMEs), mostly from the cytochrome P450 (CYP) enzyme family, might describe some apparent diversity in drug response, other non-genetic variables may also play a significant role. By considering genetic and non-genetic drug response factors, personalized medicine aims to maximize pharmacological therapy for each unique patient. One non-genetic component, inflammation, has been shown to significantly impact drug metabolism, particularly by inhibiting the cytochrome P450 (CYP450) drug-metabolizing enzymes. As a result, it influences the discrepancy between the projected medication response based on genetics and the actual phenotype [15]. 

This study focuses on the ways that inflammation affects medication metabolism. CYP3A4 seems to be the most affected by inflammation, supporting clinical data on CYP3A4 drug substrates. Evidence from in vitro models has helped to establish that various CYP isoforms have distinct susceptibilities to being downregulated by inflammatory mediators.

The inflammatory stimuli also influenced the pattern of CYP isoform downregulation. In their research, Lloret-Linares et al. [37] demonstrated the metabolic activities of CYP were not connected to the contents of their liver or intestines. Therefore, inflammation may function independently of or in addition to expression. A decrease in chronic low-grade adipose tissue inflammation does not seem to be a mechanism by which bariatric/metabolic surgery enhances metabolism in humans [41], and the subject must be assessed on a case-by-case (rather than blanket) basis. However, systemic inflammation enhancements follow significant weight loss over a period.

The investigations indicate that concurrent medication and various patient- and disease-related factors modulate the activity of the crucial CYP450 enzymes, leading to alterations in drug metabolism that were not predictable from genotype or by simply examining it as a conventional drug–drug interaction [1].

Given the number of CYP substrates among antidepressants and frequently administered co-medications in this patient group, Gloor et al. indicated that phenotyping of metabolizing enzymes before therapy begins and/or adjustment could prove cost-effective. Additional viewpoints encompass phenotyping in clinical settings and several written and simple-to-follow protocols to include phenotyping results in clinical decision-making. Creating point-of-care phenotyping assays and allowing doctors to quickly alter drug doses to each patient’s unique metabolic features may greatly increase access to personalized medicine [30].

It seems that bariatric surgery and obesity have an isozyme-specific impact on CYP-mediated metabolism. In addition, the results are very drug specific, and research design is a critical factor in evaluating the findings. It is necessary to comprehend the effects of RYGB to make dosing recommendations for this population. One cannot just address the acute consequences of RYGB-induced physiology and anatomical changes on the gastrointestinal system. Obesity is a systemic condition, and bypass surgery effectively lowers body weight. Therefore, the patient’s physiology will change over time due to quick and significant weight loss in addition to changed gastrointestinal physiology with the potential for altered medication absorption. These results led Brocks et al. to the conclusion that given the information gap, it is essential that continuously dosed drugs that have a limited therapeutic margin, including anticonvulsants and immunosuppressive treatments, be thoroughly monitored in the months after surgery [13].

## 5. Strengths and Weaknesses of the Project

Generally, the clinical utility of our approach and the observations of the present study will need to be confirmed in a larger population. These findings do not provide conclusive evidence of whether these changes are attributable to all populations.

Our extensive study provided new insights into the complex impact of bariatric surgery on the pharmacokinetics of 7 probe drugs after their simultaneous administration in obese patients.

Genotyping of the subjects would be valuable since bariatric surgery (or any other intervention under evaluation) may have variable effects in poor, extensive, and ultra-rapid metabolizers.

Although it would be desirable, collecting blood and urine samples for longer than 6 hr to cover the elimination phase of drugs is not feasible for practical reasons, such as the duration of the patient’s visit to the clinic.

The main limitation of this study is the small size of the study population coupled with a high diversity of treatments precluding a detailed analysis of single-molecule impact. This population, however, reflects the real-life pattern of patients deferred to a tertiary center and highlights the difficulty of the primary care to integrate all required information into an algorithm for phenotypic prediction. In this context, development of point-of-care phenotyping tools would considerably improve personalized pharmacological decision-making. Additional limitations include the possible omission of very rare or unidentified genetic determinants in the genotyping array as well as the intrinsic limitation of all prediction algorithm relying on the quality of the data coupling genetic variations to functional consequences for accuracy.

Table 6 demonstrate the statistically significant differences between groups.

## Figures and Tables

**Figure 1 jpm-13-01042-f001:**
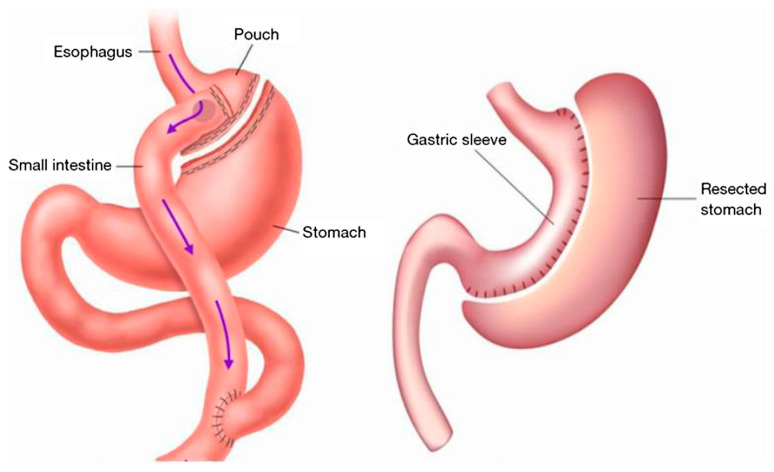
Most common types of bariatric surgeries: “Sleeve gastrectomy” and “Roux-en-Y gastric bypass” or RYGB.

**Figure 2 jpm-13-01042-f002:**
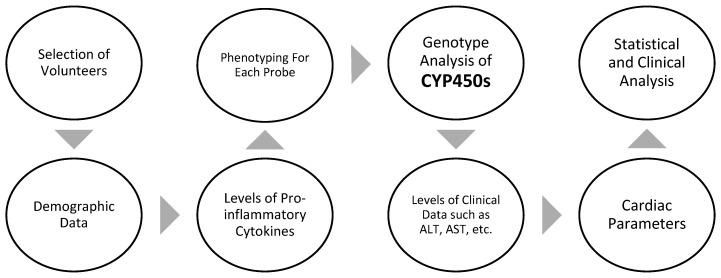
A summary of the study protocol.

**Figure 3 jpm-13-01042-f003:**
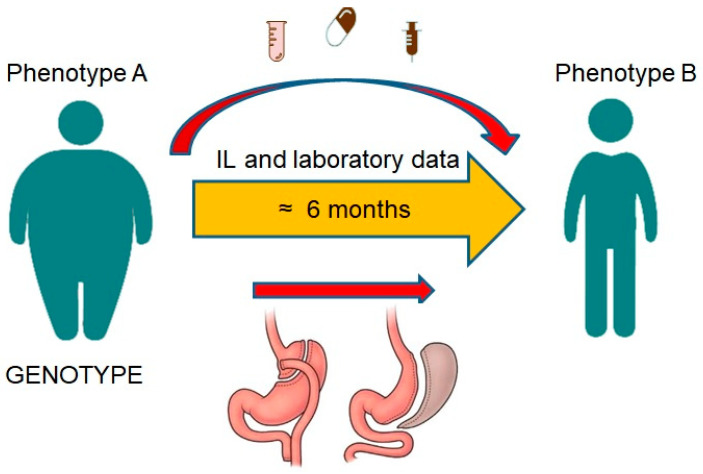
Briefly, the method of study is demonstrated.

**Figure 4 jpm-13-01042-f004:**
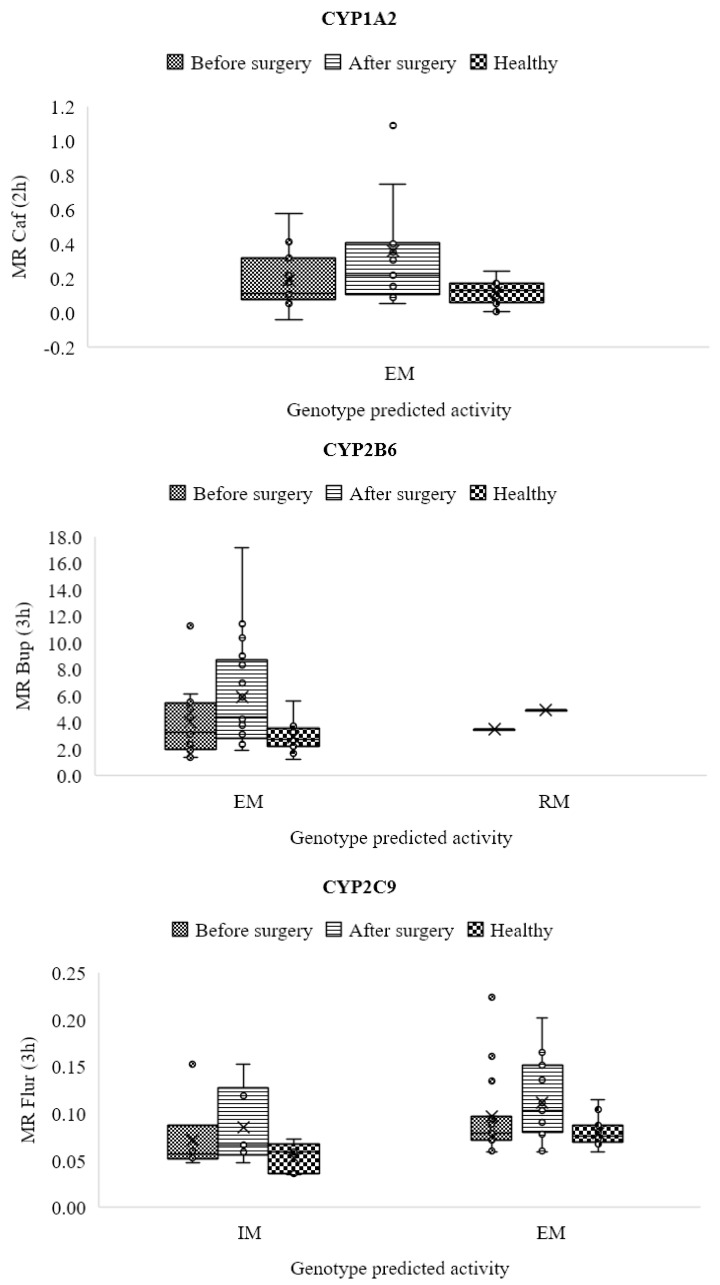
Genotype-predicted phenotypic metabolic ratios for six CYP450 isoforms in the two research groups based on genotype and phenotype results of the study population.

**Figure 5 jpm-13-01042-f005:**
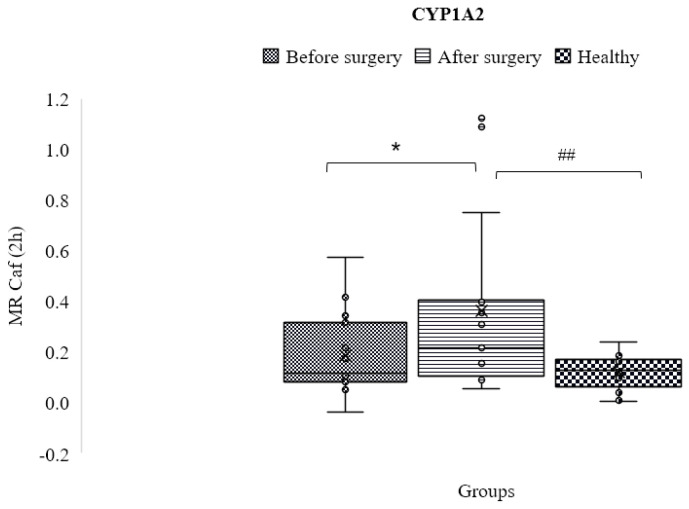
Mean metabolic ratios of caffeine at 2 h after cocktail administration in the control group as compared to obese patients before and following treatment (* and ## mean statistically significant difference).

**Figure 6 jpm-13-01042-f006:**
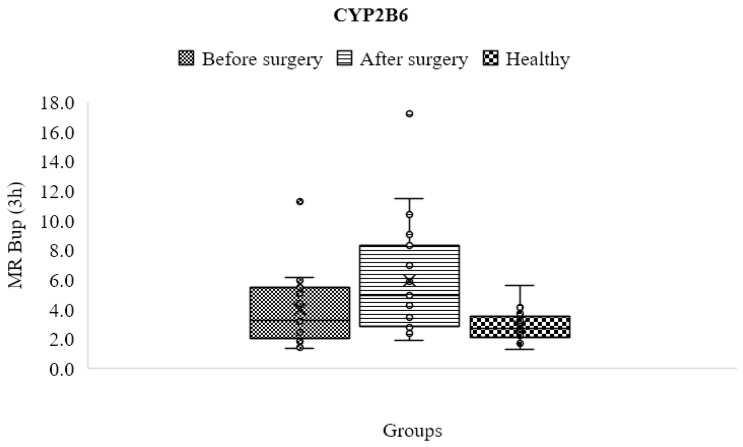
Mean metabolic ratios of bupropion at 3 h after cocktail administration in the control group compared to obese patients before and after the treatment course.

**Figure 7 jpm-13-01042-f007:**
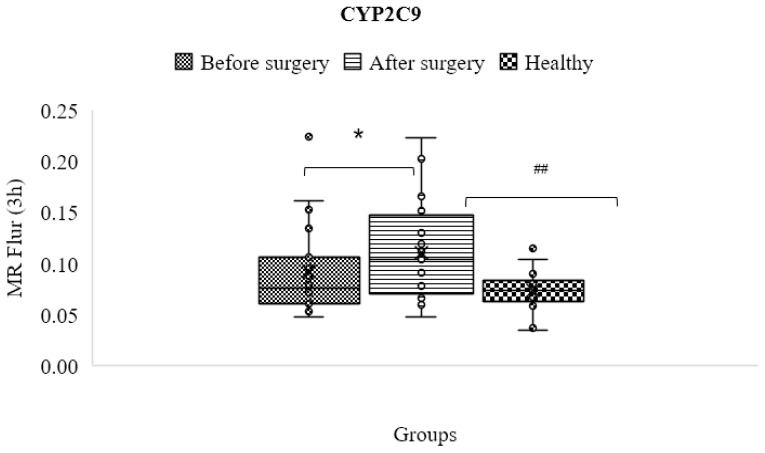
Mean metabolic ratios of flurbiprofen at 3 h after cocktail administration in the control group compared to obese patients before and after the treatment course (* and ## mean statistically significant difference).

**Figure 8 jpm-13-01042-f008:**
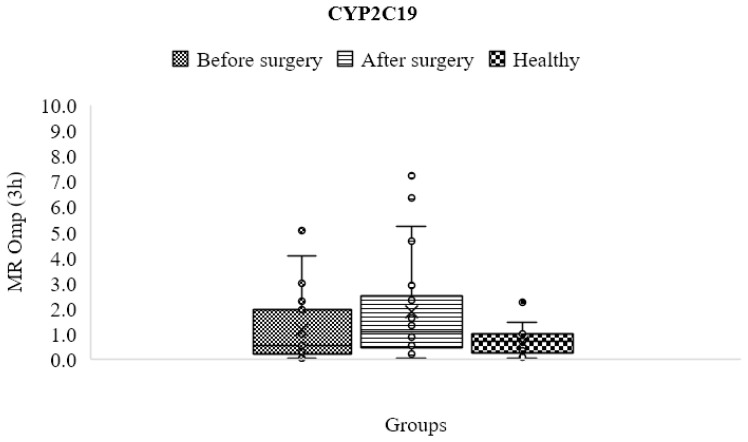
Mean metabolic ratios of omeprazole at 3 h after cocktail administration in the control group compared to obese patients before and following treatment.

**Figure 9 jpm-13-01042-f009:**
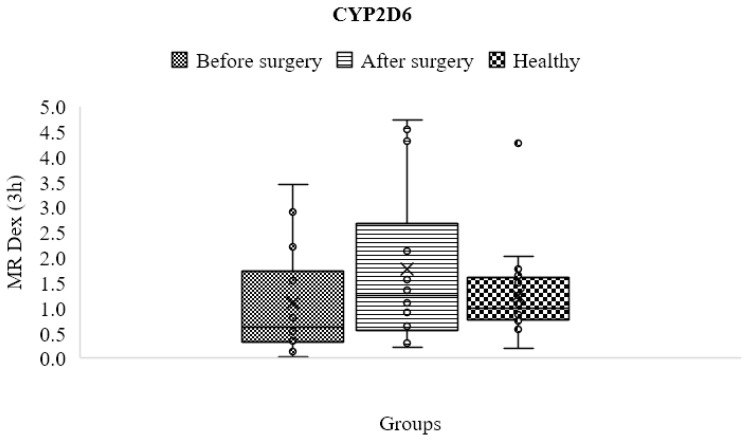
Mean metabolic ratios of dextromethorphan at three h after cocktail administration in the control group as compared to obese patients before and following treatment.

**Figure 10 jpm-13-01042-f010:**
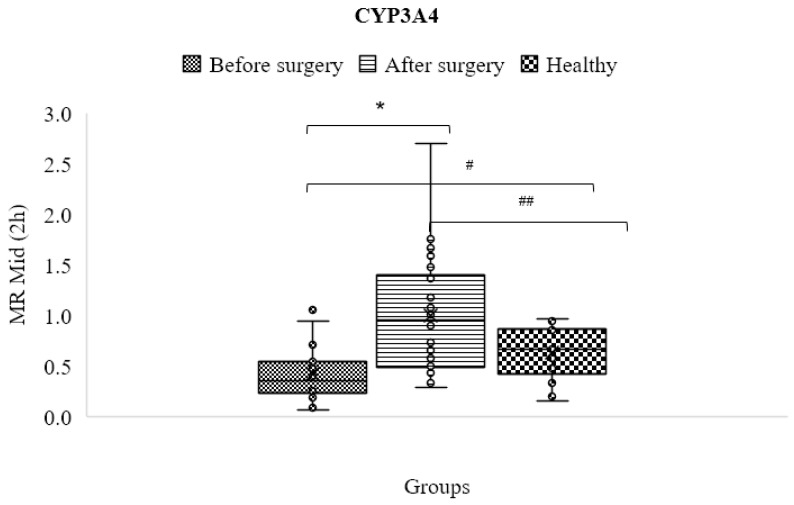
Mean metabolic ratios of midazolam at 2 h after cocktail administration in the control group compared to obese patients before and following treatment (*, # and ## mean statistical significant difference).

**Figure 11 jpm-13-01042-f011:**
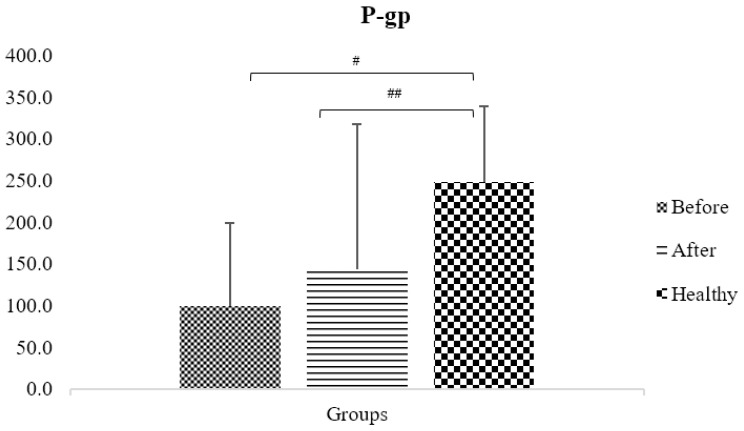
The difference between obese patients before and after three months of escalating treatment regimens, as measured by the area under the fexofenadine plasma concentration–time curve, and the control group of healthy volunteers (# and ## mean statistical significant difference).

**Table 1 jpm-13-01042-t001:** WHO classification based on BMI [10].

BMI (kg/m^2^)	Classification
<18.5	Underweight
≥18.5 and <25.0	Normal weight
≥25.0 and <30.0	Overweight
≥30.0	Obese (in general)
≥30.0 and <35.0	Obese class I (moderate obesity)
≥35.0 and <40.0	Obese class II (severe obesity) and Obese class III (morbid obesity)
≥40.0	Classification Underweight

**Table 2 jpm-13-01042-t002:** Demographic and clinical characteristics.

Parameters	Healthy Participants	Obese Patients before Treatment	Obese Patients after Treatment
No (%) of subjects	21	24	
Sex No. (%) M:F	14:7 (67:33)	3:21 (13:87)	
Age (years)	30 ± 8.9	36.6 ± 8.7	
BMI (Kg/m^2^)	24.3 ± 3.0	45.1 ± 4.2 *	32.5 ± 3.6 *
High blood pressure No. (%)	0	2 (8.3)	
Fatty liver No. (%)	0	3 (12.5)	
ESR (mm/Hr)	-	30.1 ± 15.7	20.2 ± 18.1 *
AST (U/L)	-	20.9 ± 7.8	16.8 ± 4.6 *
ALT (U/L)	-	26.6 ± 13.9	15.7 ± 5.9 *
HbA1c (%)	-	5.3 ± 0.5	5.0 ± 0.4 *
TSH (micIU/mL)	-	4.3 ± 4.1	2.4 ± 1.3
Cr (mg/dL)	-	0.9 ± 0.1	0.9 ± 0.1
Smokers No. (%)	0	12 (50)	
Alcohol cons. No. (%)	0	4 (16.7)	
SV-index (cc/m^2^)	-	31 ± 5.7	40.6 ± 5.7 *
CO-index (cc/m^2^)	-	2425 ± 475	2788 ± 663 *
IL-1β (pg/mL)	0.9 ± 0.6	2.1 ± 3.1	3.1 ± 5.2
IL-6 (pg/mL)	2.5 ± 1.5	7.3 ± 10.1	9.6 ± 11.0
Medication use, No. (%) of subjects			
Metformin	0	4 (16.7)	
Statins	1 (4.7)	2 (8.3)	
ARB	0	4 (16.7)	
CCB	0	2 (8.3)	
Β-blockers	0	4 (16.7)	
Aspirin	0	3 (12.5)	
Other NSAIDs	0	4 (16.7)	
Antidepressants	0	1 (4.2)	
PPI	1 (4.7)	4 (16.7)	
OCP	2 (9.5)	2 (8.3)	

The presentation of continuous variables is as mean ± SD. * Between research groups, there are significant differences in demographic factors (*p* < 0.05).

**Table 3 jpm-13-01042-t003:** CYP450s genotype of obese patients and healthy volunteers (control group).

CYP450	Control Group Genotype (%)	Obese Patients’ Genotype (%)
CYP1A2	Ex (100)	Ex (100)
CYP2B6	Ex (100)	Ex (95.7)-Ra (4.3)
CYP2C9	IM (26.3)-Ex (73.7)	IM (27.3)-Ex (72.7)
CYP2C19	IM (21.1)-Ex (36.8)- UR (42.1)	IM (27.3)-Ex (31.8)- UR (40.9)
CYP2D6	IM (31.6)-Ex (36.8)-UR (31.6)	IM (40)-Ex (60)
CYP3A	PM (21.1)-IM (68.4)-Ex (10.5)	PM (13.6)-IM (72.7)-Ex (13.6)

PM: poor metabolizer; IM: intermediate metabolizer; Ex: extensive metabolizer; Ra: rapid metabolizer; UR: ultra-rapid metabolizer.

**Table 4 jpm-13-01042-t004:** Mean metabolic ratios for the six CYP450 isoforms in the two experimental groups.

Isoform	Phenotypic Parameter **	Control Group (C) *	Obese- BS *	Obese- AS *	*p*-Value C vs. BS	*p*-Value C vs. AS	*p*-Value BS vs. AS
CYP1A2	C_2h paraxanthine_/C_2h caffeine_	0.129 ± 0.073	0.166 ± 0.145	0.341 ± 0.396	0.31	0.02	0.01
CYP2B6	C_3h OH-bupropion_/C_3h bupropion_	2.855 ± 0.988	3.954 ± 2.257	5.969 ± 3.703	0.05	0.001	0.01
CYP2C9	C_3h OH-flurbiprofen_/C_3h flurbiprofen_	0.074 ± 0.019	0.092 ± 0.043	0.110 ± 0.047	0.08	0.000001	0.01
CYP2C19	C_3h OH-omeprazole_/C_3h omeprazole_	0.686 ± 0.558	1.683 ± 2.767	1.887 ± 2.003	0.13	0.02	0.56
CYP2D6	C_3h dextrorphan_/C_3h dextromethorphan_	1.242 ± 0.835	1.091 ± 1.082	1.761 ± 1.592	0.65	0.22	0.15
CYP3A4/5	C_2h OH-midazolam_/C_2h midazolam_	0.633 ± 0.253	0.419 ± 0.257	1.000 ± 0.590	0.01	0.01	0.00003

* Control group: healthy, non-obese volunteers; Obese-BS: Obese patients before surgery; Obese-AS: Obese patients after surgery. ** Metabolic ratios of parent drug to metabolite plasmatic concentrations were used to calculate phenotypic indexes. Mean ± standard deviation is shown for metabolic ratios.

**Table 5 jpm-13-01042-t005:** Fexofenadine AUC0-3 after cocktail administration among the two study groups.

Pump	Phenotypic Parameter ^#^	Control Group (C)	Obese-BS *	Obese-AS *	*p*-Value C vs. BS	*p*-Value C vs. AS	*p*-Value BS vs. AS
P-gp	AUC_0-3 fexofenadine_	248 ± 91	99.9 ± 98.7	143.1 ± 174.4	5.1 × 10^−6^	0.02	0.18

# A plasmatic concentration of the parent drug (AUC0-3) was used to establish phenotypic indices. * Control group: Obese-BS: Obese patients Before Surgery; Obese-AS: Obese patients After Surgery.

**Table 6 jpm-13-01042-t006:** Statistically significant differences between groups and the guideline to translate the colors.

Significancy	CYP1A2	CYP2B6	CYP2C9	CYP2C19	CYP2D6	CYP3A4	P-gp
Before vs. After	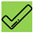	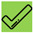	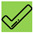	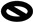	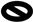	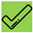	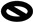
Before vs. Control	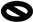	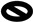	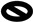	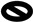	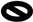	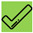	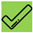
After vs. Control	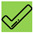	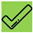	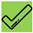	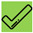	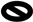	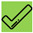	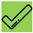

Categories	Significant (*ρ*-value < 0.05)	Non-significant (*ρ*-value > 0.05)
Before vs. After	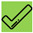	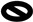
Before vs. Control	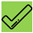	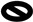
After vs. Control	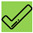	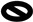

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
