# Peer review of "Impact of Obesity and Bariatric Surgery on Metabolic Enzymes and P-Glycoprotein Activity Using the Geneva Cocktail Approach"

_jpm, 2023, doi:10.3390/jpm13071042_

Round 1
Reviewer 1 Report
Ghasim et al reported about the effect of obesity and bariatric surgery on metabolic enzymes and P-glycoprotein. The manuscript was well written, however I would like to address several points:
1. The manuscript was not formatted following the template. No line number, therefore, it is hard to point out some sentences that need to be rephrased. I would suggest authors to follow JPM MDPI template for resubmission.
2. Please re-make the table 1 about the BMI classification, I am not sure if this was copied directly from WHO website or remade by authors. Use the unified font like the body text.
3. Figure one about most common types of bariatric surgeries need to be improved.
4. Figure legend, for example figure legend for Figure 2, is not sufficiently written. It does not explain about the figure. This also applied for other figures, figure legends need to be improved.
5. Result parts need an extensive improvement. The title of subchapter were not sufficiently written. Title must briefly summarize the key findings. I did not find it in all titles in result part because authors only wrote "cardiac results", "pro-inflammatory cytokines results", or "genotype results". Please rephrase the title that highlight the important findings. I would suggest to majorly revise the result parts.
6. Figure 4-11 needs to be revised. individual dots representing repetitions must be unified for each group. Using different color for each group is advisable rather than using different filling pattern. For Figure 4, please make it at one, not to be chunked into two parts. Again, figure legends for these figures need an extensive improvement. How many n or repetitions need to be written. In Figure 4, IM, EM, and UM need to be defined in figure legends.
7. Discussion part was nicely written, however extensive English editing is required.
Extensive editing is required. Proofreading by native English speaker is advisable.
Author Response
- The manuscript was not formatted following the template. No line number, therefore, it is hard to point out some sentences that need to be rephrased. I would suggest authors to follow JPM MDPI template for resubmission.
Answer 1: It has been edited as the respected reviewer explained.
- Please re-make the table 1 about the BMI classification, I am not sure if this was copied directly from WHO website or remade by authors. Use the unified font like the body text.
Answer 2: It has been edited as the respected reviewer mentioned.
- Figure one about most common types of bariatric surgeries need to be improved.
Answer 3: It has been modified as the respected reviewer asked.
- Figure legend, for example figure legend for Figure 2, is not sufficiently written. It does not explain about the figure. This also applied for other figures, figure legends need to be improved.
Answer 4: They have been edited as the respected reviewer mentioned.
- Result parts need an extensive improvement. The title of subchapter were not sufficiently written. Title must briefly summarize the key findings. I did not find it in all titles in result part because authors only wrote "cardiac results", "pro-inflammatory cytokines results", or "genotype results". Please rephrase the title that highlight the important findings. I would suggest to majorly revise the result parts.
Answer 5: It has been modified as the respected editor explained.
- Figure 4-11 needs to be revised. individual dots representing repetitions must be unified for each group. Using different color for each group is advisable rather than using different filling pattern. For Figure 4, please make it at one, not to be chunked into two parts. Again, figure legends for these figures need an extensive improvement. How many n or repetitions need to be written. In Figure 4, IM, EM, and UM need to be defined in figure legends.
Answer 6: They have been edited as much as possible.
- Discussion part was nicely written, however extensive English editing is required.
Answer 7: It has been edited by an English language expert as the respected reviewer mentioned.

Reviewer 2 Report
In this study, the metabolism of the six drug-metabolizing enzymes CYP1A2, CYP3A, CYP2B6, CYP2C9, CYP2C19, and CYP2D6 as well as the efflux transporter P-gp was determined in healthy controls, obese patients, and obese patients undergoing bariatric surgery. The effect of bariatric surgery on drug metabolism has been studied rudimentarily, which can aid in the future adjustment of drug dosage for patients undergoing bariatric surgery.
However, I still have a few minor questions regarding this article, which I trust the author will address.
1.According to some studies, the activity of CYP2D6 decreases following bariatric surgery. In this study, CYP2D6 levels are elevated. What accounts for this difference?
2.The primary sites of drug metabolism are the liver and gastrointestinal tract, as is common knowledge. The detection method described in this paper can represent the metabolic activity of these metabolic enzymes and transporters, but cannot directly detect changes in intestinal or intestinal tract expression. Studying the underlying mechanism is extremely valuable. What does the author believe about this issue?
3. The article does not indicate whether the method of bariatric surgery is SG, RYGB, or both. In my opinion, RYGB leaves a portion of the small intestine, which affects the absorption, distribution, and metabolism of medications; therefore, SG and RYGB are the only viable options. Please explain whether the effects of the two surgical procedures on drug metabolism differ.
4.Numerous variables affect drug metabolism, including age, gender, circadian rhythm, disease status, etc. Are these irrelevant variables excluded from the article?
5.The final conclusion of the article is that six months after bariatric surgery, the inflammatory state of obese patients increased compared to preoperative levels, whereas the literature cited in the article indicates that the inflammatory state will decrease drug metabolism activity. Does this contradict the findings? And in the majority of studies, the inflammatory status of patients following bariatric surgery improves. Why did the inflammatory condition of the participants in this study worsen?
I think some places in the paper still need to make appropriate grammatical modifications
Author Response
- According to some studies, the activity of CYP2D6 decreases following bariatric surgery. In this study, CYP2D6 levels are elevated. What accounts for this difference?
Answer 1: As mentioned in the manuscript, in the case of this isoenzyme, the changes were not significant and it can be emphasized that for this isoenzyme the variation was very high and the concentration of the drug and metabolite was also low, and as a result, the number of the concentration of the drug and metabolite that could be evaluated was. In the end, this is our result anyway and it shows that for many reasons, including high variation, as well as gender and ethnic differences, more studies are needed in this field with a larger sample size.
Manuscript: A slight difference existed in the dextromethorphan metabolic ratio (as CYP2D6 substrate), revealing the fact that the values of CYP2D6 MR seems not to be affected by obesity/bariatric surgery, which was similarly demonstrated by de Jong et al.; In their study CYP2D6 shows to be least sensitive to inflammation, which might be due to the fact that it is not inducible by nuclear receptors and therefore not sensitive to inflammation-induced alterations of the levels of PXR , CAR , and AhR that regulate the expression of other CYPs [15].
- The primary sites of drug metabolism are the liver and gastrointestinal tract, as is common knowledge. The detection method described in this paper can represent the metabolic activity of these metabolic enzymes and transporters, but cannot directly detect changes in intestinal or intestinal tract expression. Studying the underlying mechanism is extremely valuable. What does the author believe about this issue?
Answer 2: Yes, it is true that some isoenzymes are scattered in different parts of the body, including the intestine, liver, brain, and lungs, and the result is, especially with the hypothesis of the effect of general inflammatory factors, and it does not include only liver isoenzymes, and a result of the total activity of isoenzymes is considered. We agree with the referee, but we all agree that the most important organ involved in metabolism is the liver, and among the isoenzymes, there is almost only 3A, which has a high distribution in the intestine, and despite the fact that during surgery, a part of the intestine was removed, but in most of the cases, the activity level either increased or did not change, which again seems to be a reason for the improvement of enzyme activity following the improvement of liver function.
- The article does not indicate whether the method of bariatric surgery is SG, RYGB, or both. In my opinion, RYGB leaves a portion of the small intestine, which affects the absorption, distribution, and metabolism of medications; therefore, SG and RYGB are the only viable options. Please explain whether the effects of the two surgical procedures on drug metabolism differ.
Answer 3: As mentioned in the text about the differences between the two processes, possible consequences and mechanisms involved in the observations, it should be mentioned that the advantage of this study is the comparison of each person with himself; As the goal of personalized medicine.
Manuscript: Omeprazole MR was approximately increased (12.1 %) in patients; however, this difference did not reach statistical significance. When the AUC of parent drug was examined, not only they did not decrease after surgery (whereas the opposite would be expected due to decrease in absorption site), but also they increased significantly in patients who underwent bypass procedure (since the pH is higher in this type rather than sleeve surgery[40]). These results are most likely attributable to the fact that the entric coated omeprazole granules, in new pH had higher AUC. [7].
- Numerous variables affect drug metabolism, including age, gender, circadian rhythm, disease status, etc. Are these irrelevant variables excluded from the article?
Answer 4: We agree with the respected reviewer, and exactly because of this reason and in order to reduce the changes and effective factors, this study was designed in pairs and each patient was compared with him/herself to minimize the influence of other factors.
- The final conclusion of the article is that six months after bariatric surgery, the inflammatory state of obese patients increased compared to preoperative levels, whereas the literature cited in the article indicates that the inflammatory state will decrease drug metabolism activity. Does this contradict the findings? And in the majority of studies, the inflammatory status of patients following bariatric surgery improves. Why did the inflammatory condition of the participants in this study worsen?
Answer 5: We agree with the reviewer's opinion and even though the inflammation decreases after six months and the exact result of enzyme activity changes also confirms the same issue. But the levels of interleukins can be changed by several inflammatory factors; including stress and other factors. In particular, the result of our study has proven that the level of interleukins alone is not a suitable measure to estimate the level of inflammation in the body, especially in a long-term period (this is the case), and a better measure should be considered in this field, including CRP. As mentioned in the text:
Although improvements in systemic inflammation follow significant weight loss over many months occurred, a reduction in chronic low-grade inflammation of adipose tissue does not appear to be a mechanism by which bariatric/metabolic surgery improves metabolism in humans [41] and the subject has to be evaluated case by case (not totally).

Round 2
Reviewer 1 Report
Authors have modified the manuscript fairly, although it is not easy to follow because they did not indicate the changes (e.g. which lines, what they removed/added) in the author's reply document.
minor correction is needed